# Passion, commitment, and burnout: Experiences of Black gay men working in HIV/AIDS treatment and prevention in Atlanta, GA

**Marxavian Jones**[1]*, **Justin C. Smith**[1,2,3], **Shamia Moore**[1], **Antonio Newman**[1], **Andrés Camacho-González**[4], **Gary W. Harper**[5], **Carlos del Río**[1,6], **Sophia A. Hussen**[1,6]

**1** Hubert Department of Global Health, Emory University Rollins School of Public Health, Atlanta, Georgia, United States of America, **2** Department of Behavioral Sciences and Health Education, Emory University Rollins School of Public Health, Atlanta, Georgia, United States of America, **3** Positive Impact Health Centers, Atlanta, Georgia, United States of America, **4** Department of Pediatrics, Division of Infectious Diseases, Emory University School of Medicine, Atlanta, Georgia, United States of America, **5** Department of Health Behavior and Health Education, University of Michigan School of Public Health, Ann Arbor, Michigan, United States of America, **6** Department of Medicine, Division of Infectious Diseases, Emory University School of Medicine, Atlanta, Georgia, United States of America

* Mjone51@emory.edu

**Data Availability Statement:** Although the authors cannot make their study's data publicly available at the time of publication due to identifying

## Abstract

### Background

HIV-focused organizations, care providers and research programs often hire Black gay, bisexual and other men who have sex with men (GBMSM) in their efforts to reach highly affected communities. Due to their unique social position within and outside of organizations, Black GBMSM are ideally situated to contribute to HIV care and prevention programming targeting their own communities, but may also be at risk for stress and burnout in these settings. Despite this critical role for Black GBMSM in efforts to end the epidemic, little is known about subjective experiences of Black GBMSM who work in the HIV field.

### Methods

We conducted qualitative interviews with 19 Black GBMSM who were identified as key informants. All were working in community-based organizations, clinical or academic settings in the area of HIV prevention and treatment in Atlanta, Georgia. We used a thematic analysis approach to identify salient themes with respect to the workplace experiences of Black GBMSM as well as the role of their identities in their work in the field.

### Results

Participants discussed: (1) Shared experiences and growth; (2) Work-related stressors; (3) Worker burnout; and (4) Commitment to continue working in the HIV field. On the whole, Black GBMSM derived meaning from their work, and found their intersectional identities to be a strength in fulfilling job duties. At the same time, Black GBMSM described multiple stresses faced as they balanced their personal and professional connections to this work, while also dealing with their own challenges related to discrimination, socioeconomic status,

information present in interviews, all authors commit to make the data underlying the findings described in this study fully available without restriction to those who request the data, in compliance with the PLOS Data Availability policy. For data sets involving personally identifiable information or other sensitive data, data sharing is contingent on the data being handled appropriately by the data requester and in accordance with all applicable local requirements.

**Funding:** This work was supported by the Centers for Disease Control and Prevention's Minority AIDS Research Initiative (U01 PS005112) as well as the Emory Center for AIDS Research Initiative (P30AI050409) both presented to investigator SAH. The funders had no role in study design, data collection and analysis, decision to publish, or preparation of the manuscript. URLs: www.cfar. emory.edu, www.cdc.gov/hiv/dhap/mari.

**Competing interests:** The authors have declared that no competing interests exist.

**Abbreviations:** AIDS, Acquired Immunodeficiency Disease; ASO, AIDS-Service Organization; CBO, Community-Based Organization; EHE, Ending the HIV Epidemic; GBMSM, Gay, Bisexual, and other Men who have sex with men; HIV, Human Immunodeficiency Virus; HR, Human Resources; HPTN, HIV Prevention Trials Network; PLWH, People living with HIV; PrEP, Pre-Exposure Prophylaxis; US, United States; YBGBMSM, Young Black gay, Bisexual, and other Men who have sex with men.

and health. Participants repeatedly described sacrificing their own well-being for the greater good of their communities, highlighting contributors to burnout within and outside of the workplace.

## Conclusions

Our participants derived meaning from their work in the HIV field and were affirmed by professional interactions with other Black GBMSM. At the same time, they also faced work-related and other psychosocial stressors that predisposed them to frustration and burnout. To promote workplace equity and wellness for Black GBMSM, we share recommendations for HIV-focused organizations that employ and serve men in this demographic.

## Background

In the United States (US), Black gay, bisexual, and other men who have sex with men (GBMSM) are disproportionately affected by HIV. In 2018, Black GBMSM accounted for 37% of new HIV diagnoses among gay and bisexual men in 2018, despite only making up approximately 12% of the GBMSM population [1]. It is estimated that if current HIV incidence rates persist, one in two Black GBMSM will be diagnosed with HIV in their lifetime, compared with one in eleven White GBMSM [2]. In response to these striking disparities, HIV-focused initiatives have increasingly directed their efforts towards engaging Black GBMSM in prevention, treatment, and research. To increase effectiveness, cultural relevance, and community participation in these efforts, Black GBMSM are often hired as employees to work for community-based organizations (CBOs), healthcare organizations, academic research institutions, and government entities (e.g., public health departments) implementing HIV-related programming.

Although the importance of employing Black GBMSM to work in HIV treatment and prevention is generally accepted [3], the experiences of these men while working in the HIV field are under-explored to date. Intersectionality theory posits that Black GBMSM occupy a unique social position, shaped simultaneously by systemic racism, homonegativity, and other related social structures [4–6]. Black GBMSM may be brought to work in the HIV field in large part because of their intersectional identity, which may be seen as a strength at the hiring phase (e.g., for relatability to clients/patients, which in turn could improve recruitment and retention metrics for organizations). At the same time, Black GBMSM working in HIV-focused environments remain susceptible to discrimination and stress within and outside of the workplace, as a result of multiple oppressive forces that they face [7–11]. Black GBMSM working in HIV are therefore likely to have experiences distinct from their colleagues of other racial, gender and sexual orientation backgrounds.

Adding another potential layer of complexity, due to the epidemiology described above, Black GBMSM working in the field are more likely to be either living with HIV themselves, or having been affected by HIV impacting their friends or loved ones. Similar to race and sexual orientation, HIV serostatus can facilitate relatability with clients. At the same time, there is also reason to believe that those living with HIV might experience unique stress while working in this field. Although not previously explored specifically among Black GBMSM living with HIV in the US, research conducted in Malawi and Zambia demonstrated high rates of attrition, increases in personal illnesses, and even death among healthcare workers living with HIV [12, 13].

Workplace stress among multi-disciplinary professionals in healthcare and related fields is often framed in terms of *burnout* [14]. Burnout can be defined as a syndrome of "emotional exhaustion and cynicism" and a "tendency to evaluate oneself negatively, particularly with regard to one's work" [15]. Individuals providing service for people living with HIV (PLWH) may be at high risk for burnout due to the complex psychosocial needs and trauma experienced among this group [16, 17]. This is a critically important dynamic to explore, as burnout among service providers has the potential to adversely impact not only the well-being of those providers, but also the patients and clients who they serve. In addition to concerns for job turnover, several studies have suggested suboptimal job performance due to burnout can lead to adverse outcomes for patients/clients [12, 18].

Although not focused exclusively on Black GBMSM, several studies have documented job-related stressors among HIV workers, especially those hired in a peer capacity. For example, Hidalgo et al. examined the experiences of clinic-based outreach workers within a multi-site study aiming to engage young GBMSM of color across the HIV care continuum [19]. Eight sites hired outreach workers, most of whom identified as gay and/or living with HIV, to help engage the target population who shared those characteristics. Over the three-year study period, 57% of the outreach workers resigned or were terminated. Study staff described outreach workers' challenges adjusting to workplace environment, barriers with their own health, and difficulty maintaining professional boundaries while interacting with participants [19]. In another study focused on peers working in evaluation and data collection, peers took pride in their rapport with clients, but also struggled to ensure self-care and meet their clients' needs [20].

Relatively underexplored to date is the question of how burnout may be shaped by the identities and lived experiences of the workers in question. A related study conducted among a group of Latino GBMSM HIV/AIDS volunteers, showed that burnout was shaped by experiences within the volunteer environment, motivations for volunteering, and sense of LGBT community [21]. Those who volunteered because of their own personal experiences with HIV and/or who had a higher sense of community were less likely to burn out, while those with negative experiences volunteering were more likely to burn out. These findings highlight the need to examine experiences and motivations as a first step towards mitigating potential stress and burnout.

It is notable that much of the above literature on the HIV workforce is separate from work focused on Black GBMSM's intersectional identities. One exception is the HIV Prevention Trials Network (HPTN) 073 study, a demonstration study of HIV pre-exposure prophylaxis (PrEP) that strategically aimed to increase representation of Black GBMSM researchers in leadership roles, with a goal of increasing access to Black GBMSM communities and organizations [22]. This study, which included comprehensive cultural competency components developed by a group of Black researchers and community members, achieved a 92% retention rate for participants at 12-month follow-up. However, little is known about the personal and professional experiences of Black GBMSM investigators and staff in conducting such work. This study and others initiated important discussions around leadership and meaningful inclusivity of Black GBMSM as a way of advancing HIV prevention efforts within the Black gay community at-large, but little work has examined the duality of not only engaging, but also being self-identified members of this population of interest.

In the US, meaningful involvement of Black GBMSM is critically important to efforts such as those outlined in "Ending the HIV Epidemic: A Plan for America" [23]. Given high risk for burnout among HIV care and service providers more generally (particularly peers); as well as the multiple barriers faced by Black GBMSM more generally, it is imperative that Black GBMSM in this field are given adequate support within organizations. In order to inform such

efforts, the current study sought to explore the experiences of Black GBMSM working in HIV treatment and prevention, with a focus on understanding the role of their intersectional identities in their work.

## Methods

This paper represents a secondary analysis of data from qualitative interviews conducted to inform development of an intervention to increase social capital among young Black GBMSM (YB-GBMSM) living with HIV in Atlanta, Georgia. A full description of the study methods and the development of intervention program has been published previously (6). We conducted 28 in-depth interviews with community key informants as part of the intervention development process. Key informants were HIV service and care providers in Atlanta who were either known to the study team or recommended by our youth advisory board (YAB) based on a reputation for expertise in working with YB-GBMSM. We utilized purposive sampling strategies to recruit a diverse sample participants in terms of age, race, gender, sexual orientation, educational background, and occupation. We developed a semi-structured interview guide which assessed seven domains: (1) the participants' professional and personal interactions with YB-GBMSM; (2) the strengths of, and challenges facing YB-GBMSM in general; (3) ways in which participants' own personal identities influenced their work with YB-GBMSM; (4) diversity among YB-GBMSM; (5) collaboration among CBOs; (6) cultural competence related to services provided to YB-GBMSM; and (7) ways in which social network connections might influence engagement in care.

Verbal informed consent was obtained prior to each interview. Interviews were conducted in mutually convenient locations deemed private and adequate for proper recording without distraction. Interviews were conducted by two Black and queer-identified study team members (MJ and SM), sometimes in conjunction with a YAB member who had been trained in qualitative interviewing. Interviews were digitally recorded and transcribed by a professional service. Interviews averaged approximately one hour in length, and participants also self-administered a brief demographic survey at the end of the interview. Upon completion of the interview, participants received a $50 gift card as a token of appreciation. Although workplace treatment was not a topic explicitly included in the original interview guide, early pilot interviews featured rich discussion about the work experiences of the Black GBMSM participants, and the duality of working with and being a part of this focus population. To explore this emerging theme, the team decided to ask additional questions specifically on workplace environment (i.e., *describe your experiences working at your organization*).

### Ethical considerations

The study was approved by the Emory University Institutional Review Board and the Grady Research Oversight committee.

### Thematic analysis

Interviews were digitally audio-recorded and transcribed verbatim by a professional agency. The research team used MaxQDA 2018 (VERBI Software, Berlin, Germany) to conduct thematic analysis using the following steps. Thematic analysis is a method for identifying, analyzing, and reporting patterns or themes within qualitative data, and provides rich detail and descriptions of the data [24]. First, we created a codebook including both structural, deductive codes derived from the interview guide (e.g., "challenges facing young Black GBMSM") and inductive codes that emerged from the data (e.g., "workplace challenges"). The codebook was developed and refined through iterative discussions among the team. Subsets of transcripts

were coded by multiple coders in parallel. Weekly team meetings were held to compare and refine codes, until consensus was achieved. The team then were given coding assignments, applying appropriate codes to each. The current analysis draws upon multiple codes, including "identity in the work", "cultural competence", and "challenges". Coded text was then compared across participants, yielding the overarching themes presented.

## Results

Participants spoke at length about both challenging and rewarding aspects of working in the HIV treatment and prevention fields as Black GBMSM. We categorized discussions of their experiences into four major thematic areas: (1) Shared Experiences; (2) Work-Related Stressors; (3) Worker Burnout; and (4) Commitment to Continue Working.

### Participants

Of the 28 providers who participated in the parent study, 19 self-identified as Black GBMSM and are included in the current analysis. Participant ages ranged from age 22 to 62 (mean = 36.3, SD = 10.5) years. All participants had at least a high school diploma, and most had some postsecondary education as well. Over half worked at nonprofit CBOs/ASOs–the rest worked in clinical or research settings. We did not ask participants to disclose HIV status in the post-survey questionnaire, but six men (32%) disclosed that they were living with HIV during the course of their interview. See Table 1 for further demographic characteristics.

**Table 1. Participant demographics.**

| Age | % (n) |
|---|---|
| <30 | 31.6 (6) |
| 30–39 | 42.1 (8) |
| 40–49 | 15.8 (3) |
| >50 | 10.5 (2) |
| **Education** | |
| Some College | 21.1 (4) |
| Bachelor's Degree | 36.8 (7) |
| Advanced Degree | 42.1 (8) |
| **Organization Type** | |
| Non-Profit/CBO | 63.2 (12) |
| Academic Institute | 15.8 (3) |
| Clinical Setting | 10.5 (2) |
| Other | 10.5 (2) |
| **Years in HIV related work** | |
| 1–5 | 31.6 (6) |
| 6–10 | 31.6 (6) |
| 11–15 | 10.5 (2) |
| >16 | 26.3 (5) |
| **HIV Status** | |
| HIV Positive | 31.6 (6) |
| HIV Negative | 15.8 (3) |
| Unknown/Did not mention | 52.6 (10) |

## Shared experiences

Our participants described their Black and gay identities as assets, which made them more effective in their professions. Many described instances of using their own personal experiences to connect with, mentor and even inspire their clients. The value of shared identities and experiences for building such connections is illustrated by the quote below:

> *I am better able to meet people where they're at because of knowing at the forefront of my mind what struggles look like and what that stress looks like, um, and how it can be expressed amongst community. We hurt the people that are closest to us when we are hurting ourselves. So being able to recognize that and see it for what it is, as someone hurting and trying to sift through that to get to the love that's underneath, whether it's love for themselves or love for someone else that they lost, um, and being able to heal that, is something that I'm more able to identify because I know what it looks like to be hurting and I know what it looks like to lash out. Um, and to know what it feels like to just need someone to say, 'I see you, and I'm here whenever you're ready.' (CBO staff member, 28 years old)*

Most participants described professional interactions with Black GBMSM as mutually beneficial for both service providers and their clients. Some described their own growth and maturation through the exchange of inspiration and information with other Black GBMSM facing barriers similar to what they had previously encountered.

> *I guess, it's weird to say in the work setting that I serve the role as the person's big brother. But for the last three years I worked in a program that was just strictly designed to support Black gay men. And through that program I found a lot of, you know, learned a lot about myself and fortified myself a lot about my own HIV positivity. And because of that I was able to co-found this amazing network and. . .community-based organization. (CBO staff member, 38 years old)*

In these cases, being a Black GBMSM working in service of other Black GBMSM within the HIV context was described in a very positive light–participants utilized their identity and lived experiences to fulfill their professional responsibilities. Furthermore, self-assurance about their own intersectional identities was fortified through repeated, affirming interactions with other Black GBMSM.

## Work-Related stressors

Despite the deep and often rewarding connections that our participants felt to the work and to their clients, they discussed a range of stressors relating to work in the HIV field. The most cited stressors were a) Work-Life Imbalance, b) Tokenization and Discrimination, c) Inadequate Organizational and Professional Resources, and d) Lack of Upward Mobility.

## Work-Life imbalance

Many participants lamented a lack of healthy work-life balance in their lives. Due to deep personal commitment to the work and the close connections they felt to their clients, many struggled to set boundaries with work-related tasks, leading to encroachment of these duties upon their personal time.

> *The line between professional and personal blend with my particular type of work, um, sometimes I'm seen as a mentor. . . Because working with young, Black gay men, particularly*

*positive men, it's not transactional, it's not just about the linkage. . . . It's about being a role model, not just on the clock but off the clock as well." (CBO staff member, 33 years old)*

As alluded to above, many voiced that the needs of Black GBMSM clients often extended beyond normal business hours. Participants would frequently make themselves readily available outside of official duty hours, to ensure that clients and community members received the information and resources they needed. Of note, participants specifically pointed to their shared Black GBMSM identity and experience as the main reasons why it was more difficult to set boundaries with clients.

*I do HIV prevention counseling, emotional wellness type of counseling, motivation interviewing. We do a lot around like goal-setting and addressing barriers to care. Um, I make myself readily available. Sometimes it goes beyond intervention you know, I make myself readily available. I give out my personal cell phone number. I give my email. You know, I have clients that will ask questions in the middle of the night. Um, I make myself available because I know that there are so many barriers accessing care and then staying in the care. So I try to make myself readily available to all of my clients, particularly, my young gay Black men clients (CBO staff member, 27 years old).*

One participant gave a specific example of what some of these off-hours duties could entail:

*Just last night, at 2:30 in the morning, somebody told me that they were in the midst of swallowing Oxycodone pills. And I'm half, one eye asleep, one eye open, you know, I called them and they didn't pick up, and I called them again and they text me and said they don't want to talk, they just want to text. Okay, and I talked them down. Got them into the doctor now and was over there early this morning and made sure, you know, that they got the proper services that they need. But if anything, it's not about the finesse that it took to talk the person down, it's the fact that in the midst of a person actively trying to take their life, something in them made them reach out to me. (CBO staff member, 33 years old)*

This quote highlights two important points that relate directly to shared identities between service providers and clients: Black GBMSM service providers are often able to connect with young Black GBMSM facing extreme psychosocial challenges, and they are also frequently willing to go above and beyond expectations for typical work responsibilities. Because this work is so personal for Black GBMSM, however, many participants were unable to protect their own time and energy due to overwhelming client needs.

## Tokenism and discrimination

Participants described instances in which tokenism and discrimination within organizations negatively impacted their work experiences. Many shared feeling as though they were hired simply to promote the appearance that their organization was being culturally competent in efforts targeting Black GBMSM, a population increasingly prioritized by funders. Additionally, some participants felt that their employment was contingent upon them being knowledgeable of all Black gay experiences, without consideration of the diversity among this non-homogenous population.

*So while this career has been really, really good for me, I think it is a career that tokenizes Black gay men. I think it is the career that very few are lucky to be in because people really want to see them grow. I think many of us have positions in this field because the assumption*

*is that I'm the Black gay person that knows all Black gay things, and I'm going to hire you for this role because my grant focuses on this demographic and we feel like because you're a part of it, you must be an expert, and that's not true. (CBO staff member, 35 years old)*

Experiences of tokenism were very common among our participants. Related to this tokenism, a few participants expressed discomfort with the commodification of certain parts of their identity, at the expense of their privacy. Because their employment was perceived as inherently tied to their identity as Black GBMSM (who, in some cases, were also living with HIV) these participants felt pressured to openly share very personal information (i.e., their sexuality and/ or HIV status) as a way of engaging other Black GBMSM. A few participants were still working through their own internal processes of self-acceptance of sexuality and/or HIV status and had not disclosed widely–these participants found identity-focused aspects of the job to be particularly stressful.

Of note, some participants felt not only tokenized, but more explicitly discriminated against because of their sexuality. When such workplace discrimination occurred, participants often felt that they had no recourse, due in part to perceived alliance between human resources departments and managers, as well as state employment laws that allow for at-will termination of employees without a need to prove cause.

*Like in the workplace, in the workplace like you are subjected to so much scrutiny. . .People will see a gay man as an easy target in the workplace. . .I've been you know "type-casted", "Oh you can only do this because this is what you are and that's what we feel the program's doing." So you're kinda pimped out there in a way, in the sense. But you're also manipulated just for the aims of the program. . . for the gay community as a whole, we're type casted, we're stereo-typed, we're seen as easy targets. How we address that? We live in Georgia, that's a barrier in itself right there with some of the laws that we have for employment, it's just really antiquated and doesn't really give credence to those individuals who are marginalized in the office com-munity. (Healthcare Organization employee, 29 years old)*

As a lower-level worker who is also a Black GBMSM, he is describing how he is forced to endure discrimination based on his sexuality because he fears losing his job. This participant illustrates how identity intersects with socioeconomic position and workplace hierarchy to cause disempowerment and discouragement.

## Inadequate organizational and professional resources

Because of the close connections that they felt to their clients and communities, several partici-pants described frustration when their desire to aid Black GBMSM was limited due to inade-quate resources. Organizational limitations, often related to policies or funding, were barriers to providing the necessary level of support to clients and patients. Some participants highlighted a disconnect between funding priorities and the issues that they were directly faced with in their communities:

*I'm not sure where priorities lie with, you know, federal funding, with the federal government. I'm not sure what their, what their vision is. I just know about—as a front line staff member and on the ground level, I have to, you know, look my friends and my brothers in the eye and say, you know, this program, after ten years is done, like, and because it's grant funded, there is no, there is no kind of sun setting and it being a—and being something that we have time to really process. It's like, it's done. Like, once that date, once like March 31ˢᵗ shows up, that's it. . . Um, the red tape, the red tape has me tired. There are so many stipulations to what*

*someone can do when they're just trying to help people, it's really—that's really kind of bother-some to me. (CBO staff member, 28 years old)*

In addition to a lack of resources at the organizational level, several participants cited a lack of professional resources for Black GBMSM workers as an obstacle to efficiently providing services to others. Participants identified inadequate compensation, lack of benefits, and lackluster support from employers as stressors that reduced their ability to focus on their work.

*We're overworked and underpaid! You know. . .we deal with the same issues that the clients that we're trying to help get through the day-to-day to deal with, you know. I gotta worry about housing. I gotta worry about my bills. I gotta worry about going to the doctor. I gotta worry about all those things and oftentimes organizations don't give that support. (CBO staff member, 27 years old)*

Several participants specifically mentioned that they did not have access to affordable physical or mental healthcare in spite of their employment, and cited a need for such services to be available to help them cope with work and life-related stressors.

## Lack of upward mobility

Many of our participants highlighted limited opportunities for promotion, lack of adequate benefits (e.g., paid time off and health benefits), and lack of job security as additional examples of poor treatment in the workplace. In particular, many participants voiced resentment at a perceived lack of opportunities for training and advancement for Black GBMSM working in HIV-focused organizations. Speaking about people living with HIV being hired as peer educators or peer navigators, one participant stated the following:

*I'm a firm believer, peers are great individuals but so often we overlook peers, so often we don't promote peers. Because just because I'm living with HIV doesn't mean I don't have a Master's degree. You know what I'm saying? It doesn't mean I can't be Executive Director. But often we say, "Oh you're HIV-positive, you stay right here. You talk to these people." That's wonderful but you need, people need to hear my story of progression. (CBO staff member, 33 years old)*

Several others echoed this sentiment and discussed the frustration related to being a Black GBMSM pigeonholed in certain, often front-line, roles without clear opportunities for growth. Several participants suggested this as a prime opportunity for improvement within CBOs.

*Interviewer: So what do you feel that various organizations can do better?*

*Participant: First of, all you know, continue to hire you know, highly impactful and highly qualified young gay Black men to do this work, to work for young gay Black men–continue to do that. But foster their growth. (CBO staff member, 27 years old)*

This participant and others are again referencing hierarchy within the workplace, within which Black GBMSM are typically at the bottom of the hierarchy, working in jobs that are less secure and have less opportunity for growth. This position is described in implied contrast to other people working in the organizations, who are less likely to be Black GBMSM and/or PLWH, but who may be more likely to have leadership positions and potential for advancement.

## Worker burnout

When experienced in combination, the aforementioned challenges with work-life imbalance, tokenization, discrimination, and lack of career progression culminated in experiences of burnout. Participants related this burnout directly to de-prioritization of their own self-care as they completed tasks that fell outside of their official scope of work "to get the job done," without regard to the demands such extra duties may have on their own health needs. At times, participants said that their work became so stressful that it led to participants suffering many of the same negative physical and emotional sequelae that they were working to prevent among the clients they served. Others discussed situations in which they were already beginning to impose limitations on themselves for the sake of preventing such burnout. Unfortunately, taking these types of self-care measures and setting boundaries was often accompanied with a sense of guilt.

> *My vocation is my occupation and vice-versa. So, on a daily basis, I'm out in the community, I'm communicating with the community, I'm a part of the community. And when I come home, sometimes that burnout from dealing with multiple people in the community or just multiple personalities in general, it kind of burns you out and then you want to do more for yourself because you represent, I represent the community I serve as well. I wish I could do more because I want to do more but I don't because of burnout.*

> *(Healthcare Organization employee, 29 years old)*

## Commitment to continue working

Despite the major challenges faced by Black GBMSM in these settings, our participants were overwhelmingly committed to continuing to work in the HIV field. Some participants stated that they would "work for free" and loved their work, while also acknowledging the unique challenges that they faced.

> *A lot of us will do some great work, but we're not getting paid for it. Which granted, we're gonna do it for the community but we also need to live and thrive as well, you know? Um, that's one of my biggest things that I face. I don't have enough time to give self-care because I'm working so hard.*

> *(CBO staff member, 27 years old)*

Participants described several reasons for continuing to work in HIV treatment and prevention, with major motivating factors centering around loss of loved ones, observing the impact of HIV on their Black gay communities more broadly, and for some, personal experiences living with HIV.

> *I had friends who were being diagnosed, um, when I was in college, there weren't any real places for people to go. They were still trying to figure out what was going on. So people were dying. And I took it personal, especially when I had the opportunity myself to work in this field, I made a kind of a personal promise to all of the people that I had lost that I would continue the fight for them. Since they can't fight anymore, I'm going to fight for them to help as many people as I can not have to die to this disease. (Healthcare Organization employee, 43 years old)*

Essentially all of these participants described a similar passion and deep commitment to continued service of their Black GBMSM community; work in this area was not merely an

occupation but a mission. Participants also expressed pride in the depth of their commitment to the work and to their communities, in spite of the concomitant stressors that were described.

## Discussion

Despite a plethora of research and programming specifically targeting Black GBMSM for HIV prevention and treatment initiatives, to our knowledge, this is the first study to specifically explore the experiences of Black GBMSM as service providers working to implement these initiatives in the field. Our participants described both advantages and disadvantages related to their Black GBMSM identity in their work. On the one hand, our participants were able to perform their jobs more effectively, and found more deeper meaning in them, as a result of shared identities with the clients they served. At the same time, experiences of workplace discrimination and resource limitations also created frustration, stress, and burnout. We found that Black GBMSM working in the HIV field are at high risk for stress and burnout, but in many cases continue to work in this area due to a strong personal commitment to their communities.

Burnout in HIV is well documented, though not previously linked before to the intersectional identities of Black GBMSM workers. Despite frequent documentation of such burnout, there is a lack of evidence-based interventions for addressing burnout specifically in this field. Burnout interventions in healthcare more generally often focus at the individual level, for example, by teaching mindfulness, cognitive behavioral therapy, or other strategies to cope with stressors. While these strategies may have some incremental benefit, we agree with other scholars that true prevention and mitigation of burnout must occur at the organizational level [25]. A meta-analysis of interventions targeting physician burnout, for example, suggested significantly improved effects of organization-directed interventions when compared to individual-level physician-directed interventions [26].

At the same time, strategies to address burnout among Black GBMSM working in HIV cannot ignore power dynamics related to these intersectional identities. Within organizations, our Black GBMSM participants experienced emotional exhaustion not only because of shared identities with their participants, but also because of stressors related directly to experiences of discrimination and perceived precarity of their employment within organizations. Additionally, studies show that Black GBMSM encounter racism within predominantly white LGBT spaces, as well as homonegativity within Black communities [7, 27]. Solutions must therefore explicitly address the identity and social position of Black GBMSM, both within and outside of these organizations.

Our findings and the above considerations lead directly to several concrete recommendations for organizations with respect to mitigating stress and burnout among Black GBMSM working in HIV. The key causes of stress that we identified included lack of material and emotional support from organizations, lack of respect or even discrimination relating to workers' intersectional positions, and also unique stress related to the closeness of Black GBMSM to their clients and their work. To address the lack of material and emotional support, we suggest advocating for improved financial compensation and benefits for Black GBMSM working in this area. Many of our participants described job positions in which they felt that they were "only" being hired due to their Black gay identity, and that their positions were at a lower level with precarious job security–it is important that the value of these lived experiences is reflected in financial compensation as well as subsequent job opportunities and career development. To address disrespect and discrimination against Black GBMSM, organizations must examine their processes, undergo trainings to familiarize themselves with interlocking systems of

**Table 2. Recommendations for improving the experiences of Black GBMSM working in HIV prevention and treatment.**

1. Create job descriptions (and accompanying compensation and benefit packages) that reflect the value of employees' contributions to the project and acknowledge the critical importance of knowledge and skills gained from lived experiences.
2. Provide workers with appropriate flex time to acknowledge work performed outside of traditional office hours.
3. Provide workers with paid time off for physical and mental health breaks given the emotional intensity of the work.
4. Protect workers against implied or direct demands to self-disclose personal information regarding their sexuality, HIV status, and other aspects of their social identities and/or experiences.
5. Provide workers with appropriate and affordable healthcare options and mental health resources.
6. Require all employees to complete adequate cultural sensitivity trainings on a regular basis.
7. Provide formalized mentoring and career development opportunities.
8. Include workers in programmatic decision-making and advocate for more Black GBMSM to take on leadership roles.

oppression, and ultimately provide mechanisms for Black GBMSM to provide feedback about discrimination without fear of retaliation. Finally, to address the unique trauma and stress associated with being Black GBMSM and working in this field, it is also imperative that workers receive time and benefits (i.e., insurance) to facilitate care of their own physical and mental health. Table 2 provides a listing, based on our findings, of our recommendations for improving the experiences of Black GBMSM working in HIV with respect to workplace wellness and equity.

Importantly, Black GBMSM are not always lower-level employees and are not always employed as peers. As one strategy for dealing with the problems described in this study, a few of our participants had started their own organizations which were therefore led and primarily populated by Black GBMSM themselves. Future studies to compare and contrast experience of Black GBMSM working in organizations with different demographic makeups and leadership structures might shed some light on the contribution of various facets of the work experience to burnout (or lack thereof).

## Limitations

This study was a secondary analysis of data being collected for other purposes; our sampling strategy was therefore not designed to maximize variation around the experiences of Black GBMSM in the workplace. It therefore could be the case that our participants, who were selected due to community reputation for excellence in work with young Black GBMSM, may not represent the range of workplace experiences of Black GBMSM in the HIV field. If anything, however, we would expect that this would bias our sample towards reporting more favorable experiences–since the people we were interviewing were still working in the field and active/visible in the community. This brings up a concern that other Black GBMSM working in HIV are having experiences even more stressful than what is described here. Our study is also limited by its cross-sectional design; longitudinal interviews, had they been possible, would have provided more information about trajectory of burnout in this population over time.

## Conclusions

To achieve an end to the HIV epidemic in the US, we must focus on supporting Black GBMSM, whose perspectives and leadership are vital to the development and implementation of these efforts. This study identified several factors that contribute to stress and burnout among Black GBMSM working in the field of HIV treatment and prevention. There is a need

for meaningful capacity building, physical and mental health support, and improvement of workplace conditions to adequately support the Black GBMSM who work to improve the health of their own communities. Improving these organizational conditions will hopefully contribute to the overall health of Black GBMSM, and also strengthen their ability to provide the support needed to reduce HIV incidence among Black GBMSM in general.

## Acknowledgments

We are grateful to our nineteen participants for their candid responses and willingness to share their experiences in the workplace. Transcription services were conducted by Exceptional Transcription and Business Solutions.

## Author Contributions

**Conceptualization:** Sophia A. Hussen.

**Data curation:** Antonio Newman.

**Formal analysis:** Marxavian Jones, Shamia Moore.

**Funding acquisition:** Sophia A. Hussen.

**Investigation:** Sophia A. Hussen.

**Methodology:** Marxavian Jones, Shamia Moore, Sophia A. Hussen.

**Supervision:** Sophia A. Hussen.

**Writing – original draft:** Marxavian Jones, Shamia Moore.

**Writing – review & editing:** Marxavian Jones, Justin C. Smith, Shamia Moore, Andrés Camacho-González, Gary W. Harper, Carlos del Río, Sophia A. Hussen.

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
