## [Decision Letter · Decision Letter 0]

10 Feb 2021

PONE-D-20-33257

Passion, commitment, and burnout: Experiences of Black gay men working in HIV/AIDS treatment and prevention

PLOS ONE

Dear Dr. Jones,

Thank you for submitting your manuscript to PLOS ONE. After careful consideration, we feel that it has merit but does not fully meet PLOS ONE’s publication criteria as it currently stands. Therefore, we invite you to submit a revised version of the manuscript that addresses the points raised during the review process.

We look forward to receiving your revised manuscript.

Kind regards,

Petros Isaakidis

Academic Editor

PLOS ONE

Journal Requirements:

2. Please include your tables as part of your main manuscript and remove the individual files. Please note that supplementary tables (should remain/ be uploaded) as separate "supporting information" files

Reviewers' comments:

Reviewer's Responses to Questions

**Comments to the Author**

1. Is the manuscript technically sound, and do the data support the conclusions?

Reviewer #1: Yes

Reviewer #2: Partly

2. Has the statistical analysis been performed appropriately and rigorously? 

Reviewer #1: N/A

Reviewer #2: N/A

3. Have the authors made all data underlying the findings in their manuscript fully available?

Reviewer #1: Yes

Reviewer #2: Yes

4. Is the manuscript presented in an intelligible fashion and written in standard English?

Reviewer #1: Yes

Reviewer #2: No

5. Review Comments to the Author

Reviewer #1: Overview

It was with great interest that I read this paper, and I would like to congratulate the authors on their efforts to document a surprisingly under-explored area. I found the central focus of this manuscript to be relevant for publication, from both research and public health programming perspectives.

Prior to publication, there are a number of areas which I recommend the authors consider amendment to. These I believe these would improve the framing, content and discussion presented. I have organized these suggestions according to the main headings used throughout their paper.

Introduction

In its current form, I find that the Introduction does not adequately frame the importance of the study and is lacking a robust rationale for publication. This could be resolved through a more detailed exploration of existing research which elaborates some of the key concepts, health and social aspects that the authors seek to raise. Core concepts potentially of value to introduce include: personal identity/ies in professional/volunteer/activism work; the intersection of gender, race and HIV status; the role and importance of recognition for workforces; discrimination and tokenism in HIV work etc. Of course, the authors will need to discuss which they find most relevant to detail and illuminate for the reader.

I note that burnout is defined in the second paragraph of the Discussion, but understanding this is central to the framing of the study (Introduction), findings of the authors (Results) and exploring what these mean (Discussion).

A brief PubMed search highlighted some potentially useful titles for the authors to read, consider inclusion and use the referencing to identify other potential papers to support this research (e.g. 1) Swank & Fahs (2012) An Intersectional Analysis of Gender and Race for Sexual Minorities Who Engage in Gay and Lesbian Rights Activism or 2) Molina, Dirkes & Ramirez-Valles (2017) Burnout in HIV/AIDS Volunteers: A Socio-Cultural Analysis among Latino Gay, Bisexual Men, and Transgender People).

I suggest to conclude the Introduction section with a stronger paragraph summarizing the key gap in the literature and presenting a more detailed rationale for the study.

Methodology

Whilst the authors do reference the previously published research which details the Methodology in full, I feel that inclusion of more detail here would be useful to the reader, particularly regarding the study design, recruitment and inclusion/exclusion criteria.

Additionally, I suggest some light restructuring to improve the flow of the Methodology section. Namely, opening with an overview of the study design, detailing the study population and recruitment, description of the development and adjustments to the data collection tools, interview procedures, followed by the analysis and closing with the ethical considerations (including informed consent, remuneration/benefits and details of the protocol review).

There are conflicting opinions about where to include the overview of study participants and their demographic information. In this case I would suggest moving these five sentences at the end of the sub-section titled Participants from the Methodology to the opening section of the Results.

Results

The Results are interesting and extremely valuable. The authors have presented the four major themes in an order which is logical and understandable.

I wonder if the second theme of Work Related Stressors, warrants a minor re-ordering. The authors could consider presenting first work-life balance, second tokenism and discrimination, third organizational and professional resources, and finally upward mobility. I feel that the link between the personal identities described under work-life balance is more logically followed by tokenism and discrimination. Additionally, in the same section, I wonder if the experiences of organizations requiring/encouraging individuals to share personal information (sexuality, sexual orientation or HIV status) perhaps better illustrates inadequate organization and professional resources, rather than as currently presented as an example of tokenism and discrimination?

Whilst I am strongly supportive of the use of quotes in the Results section, bringing very human elements of this research forward, I do find the balance between the authors descriptive content and quotes to be somewhat unbalanced. Preferably, the Results section to be readable without any quotes at all, and where they are included, they complement and emphasise what has been presented by the authors. In its current form, removal of the quotes would make the Results section very challenging to follow, and there are a number of examples where the authors rely almost entirely on the quote to demonstrate the result. I would therefore strongly suggest an amended ratio, with more well elaborated paragraphs describing the authors findings, and shorter quotes of the study participants. The authors have very interesting findings to present, but I miss their voice.

Discussion

With such interesting Results, I was expecting a punchier Discussion to follow. Overall, I found that the section was challenging to navigate and lacking a coherent story. Several important issues are presented, but often in vague terms or with a strong advocacy/activist message that is not necessarily well linked to the findings and discussion presented. For example, “Burnout among service providers has important implications, not only for the well-being of those providers but also for the patients and clients who they serve.” What are these implications? Or a second example, “However, simply increasing diversity or employing Black GBMSM is clearly not sufficient to address this problem – if conditions continue as described by our participants, new EHE initiatives will continue to be faced with high rates of burnout and attrition among workers, suboptimal engagement with the most impacted communities, and ultimately, lack of progress in the fight to end HIV.” This long sentence doesn’t really get to the core of what the authors are trying to convey.

The authors have some exceptionally valuable points to raise here, and to properly present these a significant revision to the Discussion section is recommended. I would advise the authors re-consider the central points they wish to make and the logical order, then construct paragraphs/sections around these which make more tangible links between finding and the supporting/contradicting literature, arguing the relevance, and proposing the recommendation(s).

The inclusion of a table specifically detailing the authors recommendations is highly valuable. However, these recommendations need to be integrated into the relevant paragraphs of the Discussion so that clearer links between the identified results, supporting literature, discussion of relevance and the proposed recommendations can be demonstrated.

Grammar, Spelling and Formatting etc

In general I found the quality of writing to be high, and note that the authors have taken care to write this paper clearly in understandable language. There are a number of minor grammatical, spelling and formatting inconsistencies and errors that require amendment, which I expect the authors will be able to address in a revised version. In text any numbers under 10 should be written out in full (e.g. “one in two Black GBMSM will be diagnosed with…”). Stylistically, throughout the Results, the authors have a tendency towards slightly repetitive phrasing of their findings (e.g. “Most participants described…” and “Participants described…”). It might be worthwhile employing some light cosmetic editing to avoid these kinds of repetition where possible, bearing in mind that some repetition is inevitable. A few times in the paper the authors make use of a colon followed by a numbered list. Where the list is short (e.g. Results, sub-section Work-Life Imbalance) I would suggest putting this into full sentences rather than utilizing a list format.

Reviewer #2: This is an important paper, which addresses a current and important gap in the literature. I think that this paper will not only be of interest to people working in the US but also more globally. However, the paper requires major revision before it can be published. This includes good copy-editing to ensure sentence structure and readability.

A minor detail, the title should include the location and should also more closely reflect the key points of the paper.

In the introduction, the team reflect why Black gay, bisexual and other men who have sex

with men (GBMSM) are an important population in terms of the epidemiology of HIV in the US context. And in the discussion the authors try to situate this paper within the broader literature.

What is missing through the methods, and analysis and on through the discussion is a focus on the uniqueness of this of this group in relation to other people working in similar positions. What is the particular interest/ value of doing a subset analysis from within the larger study? How are the black men different or similar to the other men of colour who participated in the larger study? What in particular comes from the analysis of these data that would otherwise be missed if there was not a specific focus on the group of black men who participated. There is a discussion on "tokenism" within organizations but beyond this there is little reflection/ exploration of the interaction between being a peer, being black, being GBMSM and providing HIV services.

Did the authors explore other approaches to analysis which might allow deeper exploration of the experiences shared by the study participants? The thematic approach employed feels a bit flat - a different approach is needed to do justice to the rich and unique data that you have.

6. PLOS authors have the option to publish the peer review history of their article (what does this mean?). If published, this will include your full peer review and any attached files.

Reviewer #1: No

Reviewer #2: **Yes: **Sarah Jane Steele

---

## [Author Response · Author response to Decision Letter 0]

30 Sep 2021

Reviewer #1: Overview

It was with great interest that I read this paper, and I would like to congratulate the authors on their efforts to document a surprisingly under-explored area. I found the central focus of this manuscript to be relevant for publication, from both research and public health programming perspectives.

Prior to publication, there are a number of areas which I recommend the authors consider amendment to. These I believe these would improve the framing, content and discussion presented. I have organized these suggestions according to the main headings used throughout their paper.

Thank you for your encouragement!

Introduction

In its current form, I find that the Introduction does not adequately frame the importance of the study and is lacking a robust rationale for publication. This could be resolved through a more detailed exploration of existing research which elaborates some of the key concepts, health and social aspects that the authors seek to raise. Core concepts potentially of value to introduce include: personal identity/ies in professional/volunteer/activism work; the intersection of gender, race and HIV status; the role and importance of recognition for workforces; discrimination and tokenism in HIV work etc. Of course, the authors will need to discuss which they find most relevant to detail and illuminate for the reader.

Thank you for these very helpful suggestions. Incorporating the articles that you recommend from Swank & Fahs and Molina et al., we now situate our study within an intersectionality framework, to strengthen our justification for focusing specifically on the experiences of Black gay men as a unique group working in the HIV field. 

I note that burnout is defined in the second paragraph of the Discussion, but understanding this is central to the framing of the study (Introduction), findings of the authors (Results) and exploring what these mean (Discussion).

Thank you for this suggestion – we now introduce the concept of burnout in the introduction, so as to provide context for the results and discussion.

A brief PubMed search highlighted some potentially useful titles for the authors to read, consider inclusion and use the referencing to identify other potential papers to support this research (e.g. 1) Swank & Fahs (2012) An Intersectional Analysis of Gender and Race for Sexual Minorities Who Engage in Gay and Lesbian Rights Activism or 2) Molina, Dirkes & Ramirez-Valles (2017) Burnout in HIV/AIDS Volunteers: A Socio-Cultural Analysis among Latino Gay, Bisexual Men, and Transgender People).

I suggest to conclude the Introduction section with a stronger paragraph summarizing the key gap in the literature and presenting a more detailed rationale for the study.

We have added these references as well as this paragraph, thank you for the suggestion.

Methodology

Whilst the authors do reference the previously published research which details the Methodology in full, I feel that inclusion of more detail here would be useful to the reader, particularly regarding the study design, recruitment and inclusion/exclusion criteria.

We have added details about recruitment as requested.

Additionally, I suggest some light restructuring to improve the flow of the Methodology section. Namely, opening with an overview of the study design, detailing the study population and recruitment, description of the development and adjustments to the data collection tools, interview procedures, followed by the analysis and closing with the ethical considerations (including informed consent, remuneration/benefits and details of the protocol review).

We have added these details about ethical considerations, study population, and data collection tools.

There are conflicting opinions about where to include the overview of study participants and their demographic information. In this case I would suggest moving these five sentences at the end of the sub-section titled Participants from the Methodology to the opening section of the Results.

We have moved the demographic descriptions to the results.

Results

The Results are interesting and extremely valuable. The authors have presented the four major themes in an order which is logical and understandable.

I wonder if the second theme of Work Related Stressors, warrants a minor re-ordering. The authors could consider presenting first work-life balance, second tokenism and discrimination, third organizational and professional resources, and finally upward mobility. I feel that the link between the personal identities described under work-life balance is more logically followed by tokenism and discrimination. Additionally, in the same section, I wonder if the experiences of organizations requiring/encouraging individuals to share personal information (sexuality, sexual orientation or HIV status) perhaps better illustrates inadequate organization and professional resources, rather than as currently presented as an example of tokenism and discrimination?

We have re-ordered the presentation of findings as suggested by the reviewer. However, to the last point – we feel that the section mentioned (about having to disclose sexuality and HIV status) does align more closely with tokenism – the participants are being “used” for their identities and therefore are forced to disclose. We have added text to clarify this connection, but left it within the same subsection as it was originally.

Whilst I am strongly supportive of the use of quotes in the Results section, bringing very human elements of this research forward, I do find the balance between the authors descriptive content and quotes to be somewhat unbalanced. Preferably, the Results section to be readable without any quotes at all, and where they are included, they complement and emphasise what has been presented by the authors. In its current form, removal of the quotes would make the Results section very challenging to follow, and there are a number of examples where the authors rely almost entirely on the quote to demonstrate the result. I would therefore strongly suggest an amended ratio, with more well elaborated paragraphs describing the authors findings, and shorter quotes of the study participants. The authors have very interesting findings to present, but I miss their voice.

Thank you for this suggestion. We have added more explanatory text to the results section and believe that it would now be understandable even without the very rich quotes.

Discussion

With such interesting Results, I was expecting a punchier Discussion to follow. Overall, I found that the section was challenging to navigate and lacking a coherent story. Several important issues are presented, but often in vague terms or with a strong advocacy/activist message that is not necessarily well linked to the findings and discussion presented. For example, “Burnout among service providers has important implications, not only for the well-being of those providers but also for the patients and clients who they serve.” What are these implications? Or a second example, “However, simply increasing diversity or employing Black GBMSM is clearly not sufficient to address this problem – if conditions continue as described by our participants, new EHE initiatives will continue to be faced with high rates of burnout and attrition among workers, suboptimal engagement with the most impacted communities, and ultimately, lack of progress in the fight to end HIV.” This long sentence doesn’t really get to the core of what the authors are trying to convey.

The authors have some exceptionally valuable points to raise here, and to properly present these a significant revision to the Discussion section is recommended. I would advise the authors re-consider the central points they wish to make and the logical order, then construct paragraphs/sections around these which make more tangible links between finding and the supporting/contradicting literature, arguing the relevance, and proposing the recommendation(s).

The inclusion of a table specifically detailing the authors recommendations is highly valuable. However, these recommendations need to be integrated into the relevant paragraphs of the Discussion so that clearer links between the identified results, supporting literature, discussion of relevance and the proposed recommendations can be demonstrated.

Thank you for this feedback. We have completely restructured the discussion section and hope that it addresses your concerns. We agree that the prior version was less focus and less aligned with the recommendations presented in Table 2. The revised discussion is much more focused on solutions to burnout for this population (as presented in the Table) and how these relate to other solutions that have been discussed in other settings. We hope you will find this more cohesive and compelling.

Grammar, Spelling and Formatting etc

In general I found the quality of writing to be high, and note that the authors have taken care to write this paper clearly in understandable language. There are a number of minor grammatical, spelling and formatting inconsistencies and errors that require amendment, which I expect the authors will be able to address in a revised version. In text any numbers under 10 should be written out in full (e.g. “one in two Black GBMSM will be diagnosed with…”). Stylistically, throughout the Results, the authors have a tendency towards slightly repetitive phrasing of their findings (e.g. “Most participants described…” and “Participants described…”). It might be worthwhile employing some light cosmetic editing to avoid these kinds of repetition where possible, bearing in mind that some repetition is inevitable. A few times in the paper the authors make use of a colon followed by a numbered list. Where the list is short (e.g. Results, sub-section Work-Life Imbalance) I would suggest putting this into full sentences rather than utilizing a list format.

Thank you for catching these – we have edited the paper accordingly.

Reviewer #2: This is an important paper, which addresses a current and important gap in the literature. I think that this paper will not only be of interest to people working in the US but also more globally. However, the paper requires major revision before it can be published. This includes good copy-editing to ensure sentence structure and readability.

A minor detail, the title should include the location and should also more closely reflect the key points of the paper.

We have added the location to the title.

In the introduction, the team reflect why Black gay, bisexual and other men who have sex

with men (GBMSM) are an important population in terms of the epidemiology of HIV in the US context. And in the discussion the authors try to situate this paper within the broader literature.

What is missing through the methods, and analysis and on through the discussion is a focus on the uniqueness of this of this group in relation to other people working in similar positions. What is the particular interest/ value of doing a subset analysis from within the larger study? How are the black men different or similar to the other men of colour who participated in the larger study? What in particular comes from the analysis of these data that would otherwise be missed if there was not a specific focus on the group of black men who participated. There is a discussion on "tokenism" within organizations but beyond this there is little reflection/ exploration of the interaction between being a peer, being black, being GBMSM and providing HIV services.

We have added a discussion of intersectionality to the introduction that we hope better frames/justifies the focus on this population. Further, we have added additional interpretation and context in the results section and tried to specifically highlight the role of the Black GBMSM identity in the different findings and in contextualizing the different quotes.

Did the authors explore other approaches to analysis which might allow deeper exploration of the experiences shared by the study participants? The thematic approach employed feels a bit flat - a different approach is needed to do justice to the rich and unique data that you have.

We hope that the additional contextualization provided with the quotes helps to make the data feel less flat.

---

## [Editor Report · Decision Letter 1]

10 Feb 2021

PONE-D-20-33257

Passion, commitment, and burnout: Experiences of Black gay men working in HIV/AIDS treatment and prevention

PLOS ONE

Dear Dr. Jones,

Thank you for submitting your manuscript to PLOS ONE. After careful consideration, we feel that it has merit but does not fully meet PLOS ONE’s publication criteria as it currently stands. Therefore, we invite you to submit a revised version of the manuscript that addresses the points raised during the review process.

We look forward to receiving your revised manuscript.

Kind regards,

Petros Isaakidis

Academic Editor

PLOS ONE

Journal Requirements:

2. Please include your tables as part of your main manuscript and remove the individual files. Please note that supplementary tables (should remain/ be uploaded) as separate "supporting information" files

Reviewers' comments:

Reviewer's Responses to Questions

**Comments to the Author**

1. Is the manuscript technically sound, and do the data support the conclusions?

Reviewer #1: Yes

Reviewer #2: Partly

2. Has the statistical analysis been performed appropriately and rigorously? 

Reviewer #1: N/A

Reviewer #2: N/A

3. Have the authors made all data underlying the findings in their manuscript fully available?

Reviewer #1: Yes

Reviewer #2: Yes

4. Is the manuscript presented in an intelligible fashion and written in standard English?

Reviewer #1: Yes

Reviewer #2: No

5. Review Comments to the Author

Reviewer #1: Overview

It was with great interest that I read this paper, and I would like to congratulate the authors on their efforts to document a surprisingly under-explored area. I found the central focus of this manuscript to be relevant for publication, from both research and public health programming perspectives.

Prior to publication, there are a number of areas which I recommend the authors consider amendment to. These I believe these would improve the framing, content and discussion presented. I have organized these suggestions according to the main headings used throughout their paper.

Introduction

In its current form, I find that the Introduction does not adequately frame the importance of the study and is lacking a robust rationale for publication. This could be resolved through a more detailed exploration of existing research which elaborates some of the key concepts, health and social aspects that the authors seek to raise. Core concepts potentially of value to introduce include: personal identity/ies in professional/volunteer/activism work; the intersection of gender, race and HIV status; the role and importance of recognition for workforces; discrimination and tokenism in HIV work etc. Of course, the authors will need to discuss which they find most relevant to detail and illuminate for the reader.

I note that burnout is defined in the second paragraph of the Discussion, but understanding this is central to the framing of the study (Introduction), findings of the authors (Results) and exploring what these mean (Discussion).

A brief PubMed search highlighted some potentially useful titles for the authors to read, consider inclusion and use the referencing to identify other potential papers to support this research (e.g. 1) Swank & Fahs (2012) An Intersectional Analysis of Gender and Race for Sexual Minorities Who Engage in Gay and Lesbian Rights Activism or 2) Molina, Dirkes & Ramirez-Valles (2017) Burnout in HIV/AIDS Volunteers: A Socio-Cultural Analysis among Latino Gay, Bisexual Men, and Transgender People).

I suggest to conclude the Introduction section with a stronger paragraph summarizing the key gap in the literature and presenting a more detailed rationale for the study.

Methodology

Whilst the authors do reference the previously published research which details the Methodology in full, I feel that inclusion of more detail here would be useful to the reader, particularly regarding the study design, recruitment and inclusion/exclusion criteria.

Additionally, I suggest some light restructuring to improve the flow of the Methodology section. Namely, opening with an overview of the study design, detailing the study population and recruitment, description of the development and adjustments to the data collection tools, interview procedures, followed by the analysis and closing with the ethical considerations (including informed consent, remuneration/benefits and details of the protocol review).

There are conflicting opinions about where to include the overview of study participants and their demographic information. In this case I would suggest moving these five sentences at the end of the sub-section titled Participants from the Methodology to the opening section of the Results.

Results

The Results are interesting and extremely valuable. The authors have presented the four major themes in an order which is logical and understandable.

I wonder if the second theme of Work Related Stressors, warrants a minor re-ordering. The authors could consider presenting first work-life balance, second tokenism and discrimination, third organizational and professional resources, and finally upward mobility. I feel that the link between the personal identities described under work-life balance is more logically followed by tokenism and discrimination. Additionally, in the same section, I wonder if the experiences of organizations requiring/encouraging individuals to share personal information (sexuality, sexual orientation or HIV status) perhaps better illustrates inadequate organization and professional resources, rather than as currently presented as an example of tokenism and discrimination?

Whilst I am strongly supportive of the use of quotes in the Results section, bringing very human elements of this research forward, I do find the balance between the authors descriptive content and quotes to be somewhat unbalanced. Preferably, the Results section to be readable without any quotes at all, and where they are included, they complement and emphasise what has been presented by the authors. In its current form, removal of the quotes would make the Results section very challenging to follow, and there are a number of examples where the authors rely almost entirely on the quote to demonstrate the result. I would therefore strongly suggest an amended ratio, with more well elaborated paragraphs describing the authors findings, and shorter quotes of the study participants. The authors have very interesting findings to present, but I miss their voice.

Discussion

With such interesting Results, I was expecting a punchier Discussion to follow. Overall, I found that the section was challenging to navigate and lacking a coherent story. Several important issues are presented, but often in vague terms or with a strong advocacy/activist message that is not necessarily well linked to the findings and discussion presented. For example, “Burnout among service providers has important implications, not only for the well-being of those providers but also for the patients and clients who they serve.” What are these implications? Or a second example, “However, simply increasing diversity or employing Black GBMSM is clearly not sufficient to address this problem – if conditions continue as described by our participants, new EHE initiatives will continue to be faced with high rates of burnout and attrition among workers, suboptimal engagement with the most impacted communities, and ultimately, lack of progress in the fight to end HIV.” This long sentence doesn’t really get to the core of what the authors are trying to convey.

The authors have some exceptionally valuable points to raise here, and to properly present these a significant revision to the Discussion section is recommended. I would advise the authors re-consider the central points they wish to make and the logical order, then construct paragraphs/sections around these which make more tangible links between finding and the supporting/contradicting literature, arguing the relevance, and proposing the recommendation(s).

The inclusion of a table specifically detailing the authors recommendations is highly valuable. However, these recommendations need to be integrated into the relevant paragraphs of the Discussion so that clearer links between the identified results, supporting literature, discussion of relevance and the proposed recommendations can be demonstrated.

Grammar, Spelling and Formatting etc

In general I found the quality of writing to be high, and note that the authors have taken care to write this paper clearly in understandable language. There are a number of minor grammatical, spelling and formatting inconsistencies and errors that require amendment, which I expect the authors will be able to address in a revised version. In text any numbers under 10 should be written out in full (e.g. “one in two Black GBMSM will be diagnosed with…”). Stylistically, throughout the Results, the authors have a tendency towards slightly repetitive phrasing of their findings (e.g. “Most participants described…” and “Participants described…”). It might be worthwhile employing some light cosmetic editing to avoid these kinds of repetition where possible, bearing in mind that some repetition is inevitable. A few times in the paper the authors make use of a colon followed by a numbered list. Where the list is short (e.g. Results, sub-section Work-Life Imbalance) I would suggest putting this into full sentences rather than utilizing a list format.

Reviewer #2: This is an important paper, which addresses a current and important gap in the literature. I think that this paper will not only be of interest to people working in the US but also more globally. However, the paper requires major revision before it can be published. This includes good copy-editing to ensure sentence structure and readability.

A minor detail, the title should include the location and should also more closely reflect the key points of the paper.

In the introduction, the team reflect why Black gay, bisexual and other men who have sex

with men (GBMSM) are an important population in terms of the epidemiology of HIV in the US context. And in the discussion the authors try to situate this paper within the broader literature.

What is missing through the methods, and analysis and on through the discussion is a focus on the uniqueness of this of this group in relation to other people working in similar positions. What is the particular interest/ value of doing a subset analysis from within the larger study? How are the black men different or similar to the other men of colour who participated in the larger study? What in particular comes from the analysis of these data that would otherwise be missed if there was not a specific focus on the group of black men who participated. There is a discussion on "tokenism" within organizations but beyond this there is little reflection/ exploration of the interaction between being a peer, being black, being GBMSM and providing HIV services.

Did the authors explore other approaches to analysis which might allow deeper exploration of the experiences shared by the study participants? The thematic approach employed feels a bit flat - a different approach is needed to do justice to the rich and unique data that you have.

6. PLOS authors have the option to publish the peer review history of their article (what does this mean?). If published, this will include your full peer review and any attached files.

Reviewer #1: No

Reviewer #2: **Yes: **Sarah Jane Steele

---

## [Author Response · Author response to Decision Letter 1]

19 Jan 2022

Reviewer #1: Overview

It was with great interest that I read this paper, and I would like to congratulate the authors on their efforts to document a surprisingly under-explored area. I found the central focus of this manuscript to be relevant for publication, from both research and public health programming perspectives.

Prior to publication, there are a number of areas which I recommend the authors consider amendment to. These I believe these would improve the framing, content and discussion presented. I have organized these suggestions according to the main headings used throughout their paper.

Thank you for your encouragement!

Introduction

In its current form, I find that the Introduction does not adequately frame the importance of the study and is lacking a robust rationale for publication. This could be resolved through a more detailed exploration of existing research which elaborates some of the key concepts, health and social aspects that the authors seek to raise. Core concepts potentially of value to introduce include: personal identity/ies in professional/volunteer/activism work; the intersection of gender, race and HIV status; the role and importance of recognition for workforces; discrimination and tokenism in HIV work etc. Of course, the authors will need to discuss which they find most relevant to detail and illuminate for the reader.

Thank you for these very helpful suggestions. Incorporating the articles that you recommend from Swank & Fahs and Molina et al., we now situate our study within an intersectionality framework, to strengthen our justification for focusing specifically on the experiences of Black gay men as a unique group working in the HIV field. 

I note that burnout is defined in the second paragraph of the Discussion, but understanding this is central to the framing of the study (Introduction), findings of the authors (Results) and exploring what these mean (Discussion).

Thank you for this suggestion – we now introduce the concept of burnout in the introduction, so as to provide context for the results and discussion.

A brief PubMed search highlighted some potentially useful titles for the authors to read, consider inclusion and use the referencing to identify other potential papers to support this research (e.g. 1) Swank & Fahs (2012) An Intersectional Analysis of Gender and Race for Sexual Minorities Who Engage in Gay and Lesbian Rights Activism or 2) Molina, Dirkes & Ramirez-Valles (2017) Burnout in HIV/AIDS Volunteers: A Socio-Cultural Analysis among Latino Gay, Bisexual Men, and Transgender People).

I suggest to conclude the Introduction section with a stronger paragraph summarizing the key gap in the literature and presenting a more detailed rationale for the study.

We have added these references as well as this paragraph, thank you for the suggestion.

Methodology

Whilst the authors do reference the previously published research which details the Methodology in full, I feel that inclusion of more detail here would be useful to the reader, particularly regarding the study design, recruitment and inclusion/exclusion criteria.

We have added details about recruitment as requested.

Additionally, I suggest some light restructuring to improve the flow of the Methodology section. Namely, opening with an overview of the study design, detailing the study population and recruitment, description of the development and adjustments to the data collection tools, interview procedures, followed by the analysis and closing with the ethical considerations (including informed consent, remuneration/benefits and details of the protocol review).

We have added these details about ethical considerations, study population, and data collection tools.

There are conflicting opinions about where to include the overview of study participants and their demographic information. In this case I would suggest moving these five sentences at the end of the sub-section titled Participants from the Methodology to the opening section of the Results.

We have moved the demographic descriptions to the results.

Results

The Results are interesting and extremely valuable. The authors have presented the four major themes in an order which is logical and understandable.

I wonder if the second theme of Work Related Stressors, warrants a minor re-ordering. The authors could consider presenting first work-life balance, second tokenism and discrimination, third organizational and professional resources, and finally upward mobility. I feel that the link between the personal identities described under work-life balance is more logically followed by tokenism and discrimination. Additionally, in the same section, I wonder if the experiences of organizations requiring/encouraging individuals to share personal information (sexuality, sexual orientation or HIV status) perhaps better illustrates inadequate organization and professional resources, rather than as currently presented as an example of tokenism and discrimination?

We have re-ordered the presentation of findings as suggested by the reviewer. However, to the last point – we feel that the section mentioned (about having to disclose sexuality and HIV status) does align more closely with tokenism – the participants are being “used” for their identities and therefore are forced to disclose. We have added text to clarify this connection, but left it within the same subsection as it was originally.

Whilst I am strongly supportive of the use of quotes in the Results section, bringing very human elements of this research forward, I do find the balance between the authors descriptive content and quotes to be somewhat unbalanced. Preferably, the Results section to be readable without any quotes at all, and where they are included, they complement and emphasise what has been presented by the authors. In its current form, removal of the quotes would make the Results section very challenging to follow, and there are a number of examples where the authors rely almost entirely on the quote to demonstrate the result. I would therefore strongly suggest an amended ratio, with more well elaborated paragraphs describing the authors findings, and shorter quotes of the study participants. The authors have very interesting findings to present, but I miss their voice.

Thank you for this suggestion. We have added more explanatory text to the results section and believe that it would now be understandable even without the very rich quotes.

Discussion

With such interesting Results, I was expecting a punchier Discussion to follow. Overall, I found that the section was challenging to navigate and lacking a coherent story. Several important issues are presented, but often in vague terms or with a strong advocacy/activist message that is not necessarily well linked to the findings and discussion presented. For example, “Burnout among service providers has important implications, not only for the well-being of those providers but also for the patients and clients who they serve.” What are these implications? Or a second example, “However, simply increasing diversity or employing Black GBMSM is clearly not sufficient to address this problem – if conditions continue as described by our participants, new EHE initiatives will continue to be faced with high rates of burnout and attrition among workers, suboptimal engagement with the most impacted communities, and ultimately, lack of progress in the fight to end HIV.” This long sentence doesn’t really get to the core of what the authors are trying to convey.

The authors have some exceptionally valuable points to raise here, and to properly present these a significant revision to the Discussion section is recommended. I would advise the authors re-consider the central points they wish to make and the logical order, then construct paragraphs/sections around these which make more tangible links between finding and the supporting/contradicting literature, arguing the relevance, and proposing the recommendation(s).

The inclusion of a table specifically detailing the authors recommendations is highly valuable. However, these recommendations need to be integrated into the relevant paragraphs of the Discussion so that clearer links between the identified results, supporting literature, discussion of relevance and the proposed recommendations can be demonstrated.

Thank you for this feedback. We have completely restructured the discussion section and hope that it addresses your concerns. We agree that the prior version was less focus and less aligned with the recommendations presented in Table 2. The revised discussion is much more focused on solutions to burnout for this population (as presented in the Table) and how these relate to other solutions that have been discussed in other settings. We hope you will find this more cohesive and compelling.

Grammar, Spelling and Formatting etc

In general I found the quality of writing to be high, and note that the authors have taken care to write this paper clearly in understandable language. There are a number of minor grammatical, spelling and formatting inconsistencies and errors that require amendment, which I expect the authors will be able to address in a revised version. In text any numbers under 10 should be written out in full (e.g. “one in two Black GBMSM will be diagnosed with…”). Stylistically, throughout the Results, the authors have a tendency towards slightly repetitive phrasing of their findings (e.g. “Most participants described…” and “Participants described…”). It might be worthwhile employing some light cosmetic editing to avoid these kinds of repetition where possible, bearing in mind that some repetition is inevitable. A few times in the paper the authors make use of a colon followed by a numbered list. Where the list is short (e.g. Results, sub-section Work-Life Imbalance) I would suggest putting this into full sentences rather than utilizing a list format.

Thank you for catching these – we have edited the paper accordingly.

Reviewer #2: This is an important paper, which addresses a current and important gap in the literature. I think that this paper will not only be of interest to people working in the US but also more globally. However, the paper requires major revision before it can be published. This includes good copy-editing to ensure sentence structure and readability.

A minor detail, the title should include the location and should also more closely reflect the key points of the paper.

We have added the location to the title.

In the introduction, the team reflect why Black gay, bisexual and other men who have sex

with men (GBMSM) are an important population in terms of the epidemiology of HIV in the US context. And in the discussion the authors try to situate this paper within the broader literature.

What is missing through the methods, and analysis and on through the discussion is a focus on the uniqueness of this of this group in relation to other people working in similar positions. What is the particular interest/ value of doing a subset analysis from within the larger study? How are the black men different or similar to the other men of colour who participated in the larger study? What in particular comes from the analysis of these data that would otherwise be missed if there was not a specific focus on the group of black men who participated. There is a discussion on "tokenism" within organizations but beyond this there is little reflection/ exploration of the interaction between being a peer, being black, being GBMSM and providing HIV services.

We have added a discussion of intersectionality to the introduction that we hope better frames/justifies the focus on this population. Further, we have added additional interpretation and context in the results section and tried to specifically highlight the role of the Black GBMSM identity in the different findings and in contextualizing the different quotes.

Did the authors explore other approaches to analysis which might allow deeper exploration of the experiences shared by the study participants? The thematic approach employed feels a bit flat - a different approach is needed to do justice to the rich and unique data that you have.

We hope that the additional contextualization provided with the quotes helps to make the data feel less flat.

---

## [Editor Report · Decision Letter 2]

16 Feb 2022

Passion, commitment, and burnout: Experiences of Black gay men working in HIV/AIDS treatment and prevention in Atlanta, GA.

PONE-D-20-33257R2

Dear Dr. Jones,

We’re pleased to inform you that your manuscript has been judged scientifically suitable for publication and will be formally accepted for publication once it meets all outstanding technical requirements.

Kind regards,

Petros Isaakidis MD, PhD

Academic Editor

PLOS ONE
---

## [Editor Report · Acceptance letter]

23 Feb 2022

PONE-D-20-33257R2 

Passion, commitment, and burnout: Experiences of Black gay men working in HIV/AIDS treatment and prevention in Atlanta, GA. 

Dear Dr. Jones:

I'm pleased to inform you that your manuscript has been deemed suitable for publication in PLOS ONE. Congratulations! Your manuscript is now with our production department. 

Kind regards, 

on behalf of

Dr. Petros Isaakidis 

Academic Editor

PLOS ONE